# Development of an Intelligent Urban Water Network System

Kiran Joseph [ID], Ashok K. Sharma *[ID] and Rudi van Staden

Institute for Sustainable Industries and Liveable Cities (ISILC), Victoria University, Melbourne 3011, Australia; kiran.joseph2@live.vu.edu.au (K.J.); rudi.vanstaden@vu.edu.au (R.v.S.)
* Correspondence: ashok.sharma@vu.edu.au

**Abstract:** Water and wastewater services have been provided through centralised systems for more than a century. The operational and management approaches of the water systems face challenges induced by population growth, urbanisation, and ageing infrastructure. Recent advancements in water system engineering include the development of intelligent water networks. These intelligent networks address management and operational challenges associated with pressure and flow variations in the water network and it reduces the time for identification of pipe bursts and leakages. Research is required into the development of intelligent water networks to ensure consistent data collection and analysis that can filter and aggregate into actionable events to reduce water leakage, leakage cost, customer disruptions, and damages. Implementation of an intelligent algorithm with an integrated Supervisory Control and Data Acquisition (SCADA) system, high-efficiency smart sensors, and flow meters, including a tracking mechanism, will significantly reduce system management and operational issues and ensure improved service delivery for the community. This paper discusses the history of water systems, traditional water supply systems, need for intelligent water network, and design/development of the intelligent water networks. A framework for the intelligent water network has also been presented in this paper.

**Keywords:** intelligent water network; smart water systems; leakage detection; water pipeline burst detection; cyber–physical security; artificial intelligence; IoT; wastewater; smart water management; smart water grids; drinking water networks

## 1. Introduction

Developing an intelligent water networks has been one of the long-term aims of Water Systems Engineering. This aim is further considered important due to increased urbanisation resulting from population growth, climate change effects on water resources, and ageing infrastructure requiring efficient operation and management of systems. Figure 1 illustrates the overall key features covered in this review.

### 1.1. History of Water Networks

The history of the water network characterises cultural legacy in diverse parts of the world. It summarises an overview of the hydraulic technologies that have contributed to the advancement of present technologies in water, wastewater, and storm water systems' management.

The knowledge gathered through literature outlines the history of Greek hydraulic technologies, which began in the Bronze Age and were inherited by the Romans after over 2000 years of evolution [1]. A bore well, which dates back 6174–5921 calibrated (ca) years before the present, was the first to utilise groundwater in the Yellow River area [2,3]. The first sign of urban water supply and sewage system developed in Crete, the Aegean Islands, and the Indus Valley civilisations during 3200–1100 BC [4].

**Introduction**
- History of Water Networks
- Issues of traditional water supply systems

**Intelligent Water Network**
- Necessity of Intelligent Water Network
- Structure of Intelligent Water Network
- Conservation of Energy and Water
- Asset Management and Infrastructure Monitoring

**Benefits and Economic Feasibility of Intelligent Water Network**

**Application of technologies in developing Intelligent Water Network**
- Smart Pipe.
- Smart Water Meters.
- Pressure Sensors for Pressure Management and Leakage detection.
- Real-time simulation of water networks.
- Application of modelling, optimization techniques, and decision support systems.
- Cloud computing and SCADA.
- Geographic Information System (GIS).
- Application of artificial intelligent (AI) models for water networks management.

**Architecture of Intelligent Water Network System**

**SWOT Analysis of Water System**

**Conclusions**

**Figure 1.** Structure of Review Paper.

The key to Egyptian culture was that it never lost sight of the past [5,6]. However, this is due to the unpredictability of the Nile River floods, and grain production in the region gave order and stability in the area. The ancient Egyptians not only relied on the Nile for their survival but also regarded it to be a deific force of the universe that needed to be respected and honoured [6,7].

During this period, the first documented proof of water management was seen between 2725–2671 BC [6]. An early example was found during the early Bronze Age, in a city called Mohenjo-Daro, which was a prominent urban centre of the Indus civilisation. This planned city, which was established approximately 2450 BC, sourced water from at least 700 wells and it also featured toilets in homes, sewers in the streets, and hot baths [6].

The Mesopotamians' civilisation was close behind the King Scorpion (ca.2725–2671 BC) in terms of water management. The Mesopotamians' civilisation had access to the Euphrates and Tigris River which provided water that shaped the development of this society [6–9]. Following the large-scale hydraulic projects which were implemented between 700–200 BC, the development of various kingdoms was seen in Central China. A succession of large-scale hydraulic projects was constructed to fulfil the demands of irrigation, population expansion, city defence, and to strengthen the state's authority. Aqueducts, dams, and canals were the most common large-scale hydraulic projects throughout this time [10]. In the first and second centuries BC, Eastern countries introduced urban hydraulic techniques, with the fundamental principles of water management and hydraulic technology [11,12]. Flood protection technologies were also implemented during 1900–1700 BC [13,14]. Given the similarities in hydraulic technologies created by the Mesopotamians and Egyptians, the Minoan and Indus valley civilisations should also be considered for their contributions [14]. The Minoan, Egyptian, and Indus valley civilisations affected the water management of the Mycenaeans (ca.1600–1100 BC) and Etruscans (ca.800–100 BC) in the west, as well as ancient Indians and Chinese in the east, as they built trading links with the Greek mainland [6]. The municipal water technology and administration progressed through the aqueduct of Samos and the Peisistratus for Athens [15]. Romans significantly expanded the scale of application by constructing water projects in nearly every major

city [16]. Urban water distribution and wastewater systems date back to the Bronze Age [8], with various astounding instances from the mid-third millennium BC [17]. The province of Claudius (40–60 AD) included many unique hydraulic engineering elements and strategies involved in the construction of the Pont du Gard aqueduct [18–21] that provided water to the cities of Rome and Nemausus [18,19,22].

The mathematics of pressures and flows in pipe networks has always captured significant interest among the designers, constructors, and engineers of public water systems. Several early methods have been used to calculate the flows of pipe networks. These have ranged from graphical methods to physical analogies and, conclusively, to the use of arithmetical models. These strategies have been implemented from the start of the computer age in the 1950s. One such strategy includes The Hardy cross method [23], which looked at the application of continuity of flow and potential to solve for flows in a pipe network. Other later methods for the analysis of water networks included simultaneous nodes, simultaneous loop, simultaneous pipe, and simultaneous networks [23].

Based on a United Nations report [24], the usage of water globally has risen to higher than double the rate of population growth in the last century [25]. This may lead to nearly half of the population facing water scarcity issues by 2030 [24]. Water services in Australia are often seeking novel ways to reduce water loss and expenditures related to water infrastructure. However, droughts, rising electricity costs, and increased demand for high-priced water supply sources such as desalination make long-term sustainability concerns a hard and daily economic reality [26].

Two-thirds of the Australian population lives in the five mainland states' capital cities and the nation's capital [27]. The reasons for developing efficient water distribution systems in Australia include the high per person water demand, future climate change conditions, population growth, and ageing water infrastructure. Water and wastewater services have been provided through centralised systems for more than a century [28].

Australia has a population of 25.6 million people as of 30 June 2020, and this number is increasing, with an annual growth rate of 1.3% [29]. In Australia, the average family, comprising 3–4, people uses approximately 340 litres (L) of water per person per day [29]. In dry inland areas of Australia, the average amount of usage increases to 800 L per household [29]. Melbourne water utilities encourage residents to limit their water use in order to reach a daily average of 155 L per person per day. Climate change and global warming-related variations in weather over the years are resulting in severe rainfall and temperature events. From 1910, Australia's climate has warmed by an average of 1.44 ± 0.24 °C, resulting in an increase in the frequency of extreme hot occurrences. Since 1970, rainfall in the south-west of Australia has declined by around 16 percent from April to October. Rainfall has dropped by approximately 20% in the same region between May and July since 1970 [30]. Due to a combination of climate unpredictability and long-term warming, 2019 was Australia's hottest year on record. In the future, the global mean temperature will be 1.5 degrees Celsius over the pre-industrial baseline period (1850–1900) [31]. This high variation in climate requires improved management of water resources and systems. It is expected that other developed countries will have similar justifications for promoting efficient urban municipal water systems.

### 1.2. Issues of Traditional Water Supply Systems

The main limitation of the traditional water supply system is the deterioration of the water pipe network, resulting in water losses through leakages and bursts [32]. This is because of ageing infrastructure. There are challenges with quantifying losses through leakages, identifying the location of leakages, and time delays in responding to these events. Water loss mitigation strategies are adopted by water service providers [32]. The need to balance human and environmental water requirements while protecting essential ecosystem functions and biodiversity is a recurring challenge in managing water security [33,34]. Water security concerns may be classified into the following categories: (1) direct risks to drinking water supply systems, such as terrorist attacks, earthquakes, storms, and flooding;

(2) scarcity of water resources, including the effects on economic growth and development; (3) threats to water-related ecosystems from the point and non-point pollution sources, including excessive water use, which leads to an increased usage of ecosystem resources and loss of biodiversity, as well as the human effect on ecosystems; (4) impact of climate change on increasing hydrological variability, such as increased amplitude and frequency of droughts and floods [33,34]. However, the average rate of system rehabilitation and upgrading does not keep up with expanding needs, quality demands, and ageing infrastructures. There is widespread agreement that metropolitan water systems are vulnerable to artificial and natural threats and calamities, such as droughts, earthquakes, climate change, and terrorist attacks [34].

## 2. Intelligent Water Network

Rapid population growth and urbanisation contribute to the depletion of water resources [35–37]. As a part of efforts to mitigate climate change, it has become increasingly important to adopt water management systems that have minimal environmental effects, such as reducing greenhouse gas (GHG) emissions [38]. The water industry faces new challenges regarding the sustainable management of urban water systems. There are many external factors, such as climate change, drought, and population growth in urban centres associated with water management. These factors make it more difficult to adopt a more sustainable management of the water sector [39]. Ageing infrastructure and its associated costs, monitoring of non-revenue water (NRW), and a clear understanding of the customer's demand for fair rates are some of the main challenges that water management faces [40]. Water management is required because of the increasing population and the concentration of water needs. Using advanced technologies and the adoption of more robust management models are therefore necessary to better meet the water demands [41]. The past few decades have seen an increase in water demand. This has led to increased risks of polluting water supplies and severe water scarcity in many parts of the world [42]. In several countries, the importance of water in sustainable development has been increasingly recognised; however, the water resource management and the provision of water services continue to be generally minor in the scale of public perception and government priorities. Water transfer between basins, desalination, wastewater regeneration/reuse, and well exploration are currently used to satisfy this lack of water resources [43]. More efficient water management, the water-energy nexus, as well as pressure management and smart devices would lead to a sustainable water sector [43].

The Intelligent Water Network (IWN) can be considered as a system that is informed about likely events or water network behaviours prior to their occurrence or immediately after their occurrence, and then being able to plan for, and mitigate, some of the possible outcomes or even prevent their eventuality [44]. These networks can predict system behaviours in advance or at their occurrence, including the location of such occurrence.

Currently, Australia's water authorities are dealing with unprecedented population growth, drought risk, and ageing infrastructures [45]. It is therefore essential to develop an effective water network system, such as an IWN. As a result, in case of an IWN, the asset management procedures can be planned for in these events and mitigate potential practical repercussions or prevent them completely [44]. Similar efforts are currently being undertaken by other water utilities in the developing world.

### 2.1. Necessity of an Intelligent Water Network

An IWN is needed in the modern water distribution system. It is essential because it provides an improved service to the community, manages the network optimally, reduces the breakages and losses in the network, provides a sustainable operational network, incorporates digital technology to the system to help in automation for self-decision-making of the system, improves the worth of water services to the community, reduces service delivery risks in an increasingly dynamic world with rising demands on the current assets, and proves the efficacy and reliability of the system [44,46,47]. The water sector faces

challenges in maintaining reliable and secure service provision with ageing infrastructure, urban growth, and customer financial capacity constraints. As part of an ongoing effort, research is being conducted to develop "Intelligent Networks" to meet these challenges [48].

The parameters used in developing an IWN are system elements covering pipes, reservoirs, storage tanks, and pumps; water quantity parameters, including flow rate, volume, pressure, storage capacity and levels, and water quality parameters are used to monitor the supply of water to ensure potable standards. Other parameters associated with climate can be incorporated, which include temperature, precipitation, and evaporation [26]. The IWNs extend the network asset life, defer or eliminate the need for system augmentation, reduce the operational expenditure, and minimise the impact of system failures on communities and the environment [48].

The new challenges in water management and water development planning should be aligned considering the Sustainable Development Goals (SDG), and thus there is a need to develop sustainable tools incorporating optimisation functions [49]. Since access to safe water is vital, realising the SDG may be difficult for developing economies without addressing the extension of piped water distribution networks [50]. Moreover, study should be conducted on investigating metrics regarding their applicability to SDGs for water servicing [51], which may also require life cycle costing (LCC) and life cycle assessment (LCA) analyses [35]. Even artificial intelligence (AI) can be used to compute and predict Water Quality Index (WQI) in a water supply network online [52].

### 2.2. Structure of Intelligent Water Network

IWN can have a significant role in the water industry [44]. Thompson et al. [53] described that IWN should include: (1) digital metering solutions; (2) water quality and pressure sensors; (3) use of big data systems; (4) decision support systems; (5) optimisation of assets and solutions.

The IWN model framework relies on several advanced implementations of intelligent systems developed by industries, which are discussed briefly as follows. Chauhan et al. [54] presented a Supervisory Control and Data Acquisition (SCADA) system for uninterruptible power supply by running power generating units as per the demand. The benefit of using intelligence to regulate the entire production process is that it automatically shifts the operation to meet the requirements. When a malfunction or damage occurs, the spare unit replaces the damaged unit. If more production is required, the spare unit collaborates with the other two central units. This technique also lowers the plant's operating and maintenance expenditure. Chauhan et al. [55] developed a SCADA system to maintain pH in a water treatment system. The real-time simulation of water distribution networks is based on a digital analysis system powered by real-time flow rate and pressure data. Pipeline leak detection using an Artifical Neural Network (ANN) [56] relies on linear pipelines without considering other pipelines. However, the results demonstrate its ability and credibility to detect pipeline leaks. It can detect leaks in pipes by studying pressure fluctuations. To find the precise location of a leak point, the weighted average localisation algorithm has been introduced. This hybrid ANN approach significantly improves the accuracy of pipeline leak detection based on computational results and minimises the dilemma of complicated decision-making.

A two-step burst detection and localisation approach for a long-distance water transportation system was presented to identify a pipe rupture, and the Dempster–Shafer theory [57] was applied initially. A hydraulic model was then used to calculate the location of the bursting point, which was determined to be within acceptable accuracy limits [57].

### 2.3. Conservation of Energy and Water

To reduce the greenhouse gas emissions into the environment, water utilities maintain a relatively high pumping water usage at low levels to meet the carbon-constrained future. Some water authorities already use off-peak energy metering to perform much of the high-energy long-distance pumping, with the knowledge of system pressures, flows, and

levels at any place and time through smart flow meters and pressure sensors [44]. An Internet of Things (IoT)-based water monitoring system has proven to be a huge success, as the company saved a large amount of water and has returned its investment in less than six months. Other food firms might simply copy the IoT-based water monitoring system, according to the case study. Finally, more organisations embracing IoT-based technologies to increase resource efficiency would be interesting to observe [58]; these may include water monitoring systems, processes, and areas where IoT can save significant amounts of water. Because of the adoption of the system, the amount of water (in litres) required to make one litre of beverage decreased from 2.10 to 1.96 [59]. This reduced overall annual water consumption by 6.66%, to provide an understanding of water efficient management practices that are implemented through the application of IoT. The benefits of adopting these practices are also largely expected [59].

The importance of water management within smart cities is increasingly appreciated when financial and environmental sustainability is considered in the water sector. The benefits of developing smart water grids incorporating control measures can be realised where pressure control is one of the main factors in reducing leakage [60]. Smart water systems promote more sustainable water services, reduce financial losses, and enable innovative business models that serve both urban and rural populations better [61]. Smart water grids can save money and energy by conserving water while improving customer service by increasing efficiency. Through wireless data transmission, customers can analyse their water consumption and reduce their bill, sometimes by over 30% [62].

*2.4. Asset Management and Infrastructure Monitoring*

Marney and Sharma [44] stated that intelligent networks can reduce urban water service providers' construction and maintenance costs and increase the return on asset management investments in the long term. For example, water infrastructure monitoring can provide a water authority with real-time knowledge and warnings about leaks and other problems in its water distribution infrastructure, allowing it to gain greater network control. IWN detects, alerts, and identifies network events such as leaks, bursts, and other inefficiencies using existing meter and sensor readings and sophisticated algorithms [63]. Direct measurements of the conditions of buried pipes are often impractical. Marney and Sharma [44], for this reason, suggested the use of in-pipe data collection systems for infrastructure monitoring, which would be an autonomous vehicle that roams the pipe network, or fixed cables that are permanently within the pipe network. These data are in the form of visual images of pipes and allows for precise measurements of the internal conditions of pipes. This will replace the current error-prone and relatively costly human-driven assessment systems [44].

In one study, variables such as the number of customers with a set of proposed indicators for water usage, energy consumption, glasshouse gas emissions, and necessary economic investment per capita were evaluated [49]. The information enabled water managers to classify their water systems using sustainable criteria that quantify how well certain targets are met as per the UN's Sustainable Development Goals (SDGs) [49]. For improving access to drinking water, it is crucial to extend piped water distribution networks [50]. Efficiency and sustainability are two of the main challenges facing the water sector. Increasing urbanisation and industrialisation, as well as water scarcity, are both consequences of climate change and demand [64]. Intelligent water management aims to exploit water at the regional or city level in a sustainable and self-sufficient manner. Innovative technologies, such as information and control technologies, are used to carry out this exploitation [65]. Thus, water management contributes to leakage reduction, water quality assurance, improved customer experience, and operational optimisation, among other benefits [66,67].

Application of "Information and Communication Technologies" (ICT) provides a better quality of life for consumers, and allows better management of energy and water resources [68]. There is a need to recognise that technological advancements also

promote socio-economic development [69,70] and to identify the areas requiring further development [71]. Smart water management has many advantages, including a better understanding of the water system, detection of leaks, conservation, and monitoring of the water quality. Using smart water systems, public services companies can create a comprehensive database of areas where water leaks or illegal connections occur [60].

### 3. Benefits and Economic Feasibility of Intelligent Water Network

There is a need for an IWN for its socio-economic benefits. Limited attempts [44] have been made to conduct the economic feasibility of intelligent water networks. The following should be considered for the economic assessment of IWNs [44]:

- How would remote technologies reduce the intensity of water quality monitoring?
- What are the benefits to the life-span of infrastructures due to improved real-time asset condition knowledge?
- How will the cost of these technologies affect their uptake?

From the functional considerations, an intelligent water distribution system is a traditional water distribution system with an additional layer of technologies, for use in operational and data collection/ analysis considerations, for informed decision making. It is expected that the savings from the operation of a smart water delivery system would cover technology implementation costs [72]. Higher initial costs and a lack of economic benefits are two major drawbacks. The management and analysis of data would require additional resources.

Pimpri-Chinchwad Municipal Corporation (PCMC), Pune, India suggested using intelligent technologies such as smart metering and SCADA to upgrade their current water distribution system to a continual pressurised water distribution system [73]. Barate et al. [73] predicted that the existing non-revenue water (NRW) loss of 40% can be decreased to 15%. The key benefits from IWN are savings in operating costs, reductions in bulk water supplied, deferred infrastructure augmentation, increased revenues from more efficient meter reads, improved customer relations, and reductions in non-revenue water losses [74]. Consumer satisfaction, community acceptability, and greater customer engagement and trust are difficult to quantify, but these are direct or indirect advantages of smart metering programmes [74,75].

Low-cost, efficient pipe leak detection optimises system efficiency [76], and pressure control is used in water distribution systems to reduce leaks [77]. In a case study of the introduction of an intelligent water network by a Melbourne water utility, Yarra Valley Water, the trial findings proved the system's ability to identify, classify, and detect various system events, as well as showing the potential to save water, time, and costs. The system could detect bursts 1.5 h before customers noticed, detect various forms of meter failures, locate leaks down to 25% of a distribution zone, and conserve more water [53]. The identify–localise–pinpoint strategy is a novel technique for defining the leak detection steps [78]. Early leak detection can prevent large gas spills, water leaking into the soil beneath highways resulting in sinkholes, reduce infrastructure damage, prevent damage to the surrounding environment or employees, and increase cost savings [78]. Upgrading traditional water and wastewater systems to innovative water systems or intelligent water networks can improve the overall efficiency of the system and its economic feasibility. Such a system is feasible and efficient when the calculated payback period for the high initial cost is within an acceptable range [73,79,80].

### 4. Application of Technologies in Developing Intelligent Water Networks

This section describes various technologies used in the development of IWN. IWN systems are widely used to improve efficiency and to face the water industry's challenges. Various analysis approaches and methods are used in developing IWN systems. An approach for the emergency management of Water Distribution Systems based on nodal demand control was proposed [46]. With a pipe failure, this method is used to ensure essential nodes have an appropriate head and, therefore, flow rate during an emergency.

The proposed approach would manage the delivered flow rate using a Pressure Driven Analysis method. This was based on operating control of valves and identifying the nodes where the pressure control should be implemented. The approach was based on sensitivity matrices and sensitive network nodes; the outcomes were dependent on network complexity [47]. Calibration methodologies involved using physically based knowledge, supplied by enhanced hydraulic modelling, which included pressure-dependent components of water needs in mass balance equations [81]. It takes advantage of the technological discovery that the rate of pipe failure grows as the rate of leakage increases. It is essential to adopt specific technological discoveries on pressure and flow monitoring in a water network that may be utilised to calibrate a proposed design to aid in initial leakage monitoring. IWN research should be extended where technologies could improve asset management, identify potential condition monitoring techniques, as well as promote the use of technologies to make informed economic decisions [48]. The water infrastructure needs to be met with both supply and demand challenges [82,83]. Water demand will rise in the future, causing rapid action on resource advancement, demand reduction, and increased treatment and transmission efficiency, further promoting the need for intelligent networks. The fast unfolding of communicating devices at required intervals and their application requires coordination. It is noted that the current situation in the water sector can be highlighted by a low level of maturity concerning standardisation of information and communication technologies (ICT) solutions and business processes, resulting in slow uptake of these solutions [84]. Information systems will assist decision makers in achieving targets [85], and the impact of new technologies covering sensors on decision support systems will be significant and stronger if implemented correctly [86]. The implementation of various technologies is described in this section.

### 4.1. Smart Pipe

A smart pipe is designed as a module unit with a monitoring capability that can be expanded for future sensors [87]. Real-time monitoring of several smart pipes installed in critical sections of a public water system detects flow, pressure, leaks, and water quality, without changing the hydraulic circuit's operating conditions. Low energy consumption of the wireless sensor allows it to remain operational for long periods [88].

### 4.2. Smart Water Meters

Smart Water Meters (SWMs) offer water utilities the ability to conserve water by detecting leaks early and recognising patterns in water consumption [89]. Key drivers implementing smart water meters include improved engagement with water consumers, enhanced water infrastructure planning, potentially deferring and augmenting some investments, improved peak demand forecasting and management, reduced manual meter reading, and reduced operating costs [74,79]. As SWM technology advances, fully integrated ultrasonic smart water meters with built-in communication systems are already available in Australia and other parts of the developing world [80]. With the recent introduction of high-resolution smart water meters, a new novel method for leveraging the continuous 'big data' generated by these meter fleets to create water demand curves suitable for network models have emerged [90]. A real-time monitoring system for tracking water consumption is needed to identify wastage and find opportunities to reduce consumption. The Internet of Things (IoT) is a prominent technology and is attracting attention from a broad range of industries [91]. Smart sensors and meters could be used in supply systems to provide detailed information on water consumption through transparency and visibility [36,92]. Water consumption data can be gathered in real-time and analysed to improve water-aware decision-making. Flow rate meters provide information on real-time water use, as well as identifying hotspots and showing proper management, along with predicting future consumption levels [58].

Implementation of smart water meters has shown opportunities for water-saving, which include providing consumption feedback and taking the interventions for leakage

repair. These meters have in-built communication systems such as the Narrow band Internet of Things, and they are the next generation of intelligent water meters [80].

Household smart water metering; a smart meter tracks consumption and transmits it at a specific frequency. Developing an efficient water management system requires monitoring the water system with sensors and/or actuators [93], therefore, providing instant access to customers and management entities. To manage this information, the water companies have installed advanced metering infrastructure (AMI). This will enhance hydraulic efficiency and enable leakage detection and illegal connections [94]. The Wide Bay Water Corporation intelligent metering project, which started about a decade ago, is the first large-scale implementation of Smart Water Meter technology [95]. Hence, smart water meters advance the engagement between water utilities and customers. Melbourne Water service providers are involved in a joint digital metering program to relay water usage information every day for improved water use awareness. These meters will assist providers by identifying leaks and bursts in the water network, as well as on customers' properties, resulting in water savings and economic gains. In the city of Pimpri Chinchwad, India, the increase in revenue over the 15 years from 2017 to 2031 is expected to be 436.49 hundred thousand USD [73], due to residential metering.

### 4.3. Pressure Sensors for Pressure Management and Leakage Detection

Information on water pressure and its management in the water network is essential to reduce leakages and pipe failures, including the provision of services at required pressures to meet customer service obligations. Implementation of pressure sensors at critical locations can play a crucial role in managing pressures in pipes with live information. One of the major challenges in water transportation pipes is leakage, resulting in water resource loss, human injury, and environmental harm. Much research has been published in the literature that focuses on detecting and locating leaks in water pipeline systems [96]. Pipeline transport for resources is widely used worldwide for various reasons, including its ease of design and execution and cheap operational costs [97]. Leak location detection methods use measurements of pressures and/or flows of water measured by sensors installed permanently in specific sections of the water supply system [98].

It is necessary to develop a method to detect leakages in water networks using smart pressure sensors and flowmeters. Leaks that are undetected cause financial losses and increase the maintenance costs of water supply networks [99]. As a result, it is essential to improve water supply network monitoring and diagnostic systems to identify and locate minor leaks as soon as possible [99].

Kayaalp, Zengin, Kara, and Zavrak [96] implemented a water pipeline leak detection and localisation framework using a wireless sensor network. The pressure sensors are used to monitor the water pressure through pipes and the data are obtained in real-time from the pipelines and analysed using multi-label learning methods. Sensor-driven systems work by providing knowledge and real-time data to facilitate informed decision-making by service providers and regulators [100]. There are also network-based real-time monitoring systems that use pressure data to identify and locate breaches on various water pipes.

### 4.4. Real-Time Simulation of Water Networks

The data are captured in real-time utilising Open Platform Communication Technologies [101]. Real-time simulation frameworks [101] are being developed using SCADA and Open Platform Communication interface technology for the efficient management of water distribution networks. The ongoing information from intelligent sensors and flow meters is used in such simulations and has started a new avenue for the creation of an IWN [102].

### 4.5. Application of Modelling, Optimisation Techniques, and Decision Support Systems

It is a framework that measures performance based on a set of relevant indicators and data applications and interfaces to support the decision of the managing entities, and which allows the interested parties to evaluate, build trust and confidence, and monitor

the improvements [103–105]. Models such as Epanet [106] or WaterGems [107] focus on simulations. Such tools can be supported by optimisation techniques. Among the programming models available are simulated annealing [108], fuzzy linear programming [109], and multi-objective genetic algorithms in real-time [110].

### 4.6. Cloud Computing and SCADA

These refer to the interconnection of computers and servers, through the internet, and the use of memory and storage capacity. In its most simple form, cloud computing is "a novel style of computing in which resources are dynamically scalable and virtualised being provided as a service over the Internet" [111]. Most public water services use SCADA systems for their monitoring, control, and data management [112,113].

### 4.7. Geographic Information System (GIS)

GIS plays an important role in smart water management, providing a list of the components and a spatial description. GIS is vital for the management of water systems, since it allows for the incorporation of spatial components into an oriented model, thus improving planning and management of the system. The main advantage of GIS is that it builds a simulation of reality based on data systems built to collect, store, receive, share, manipulate, analyse, and present information that is spatially referenced [114,115].

### 4.8. Application of Artificial Intelligent (AI) Models for Water Network Management

AI techniques are becoming an integral part of IWN and have contributed to water infrastructure management through different concepts, processes, and models. Innovation by water authorities can produce enhanced performance by taking advantage of the digital technology revolution. Water utilities can utilise knowledge and data to make improved decisions while enhancing service delivery and reducing costs, by using AI algorithms and big data analytics [95]. AI algorithms use several approaches, which include but are not limited to: Artificial Neural Network (ANN), Fuzzy Inference System, Neuro-Fuzzy System (NFS), Genetic Algorithms, Support Vector Machine (SVM), K-mean Clustering, etc. [102]. AI based models are used for water network assessment of pipe leakage and breakage and water contamination, including management of the system with sensor technologies [102].

Various AI approaches, such as ANN, NFS, and SVM, are used for pipe break failure prediction rate, pipe reliability evaluation, and comprehensive assessment of the IWN system [116]. Bubtiena et al. introduced a method for establishing ANN models of pipe breaks from which rehabilitation strategies, such as proactive maintenance strategies and prioritisation of its implementation, can be determined [117]. Based on historical condition observations and inspection reports, intelligent models can forecast the condition and performance evaluation of pipelines [118].

The Extreme Learning Machine developed is a novel failure rate prediction model to provide key information for optimum ongoing maintenance and rehabilitation of a water network [119]. Asnaashari et al. reported that pipe failures are becoming more common because of poor maintenance and the ageing of water treatment systems, and ANN modelling can successfully assign repair/ replacement preferences [120]. Even the water quality failure estimation, using water contamination prediction modelling tools, can be achieved with the help of AI model implementation for water supply systems [102,121–123].

## 5. Architecture for Intelligent Water Network System

Jagtap et al. developed an IoT-based water monitoring architecture for improving water efficiency in the beverage industry with the Internet of Things [58,59]. The architecture included layers for water-sensing devices, an IoT gateway, data centre for processing, and user application centre for continuous monitoring. Figure 2 shows the IoT-based communication and sensing layers, based on Jagtap et al. [58,59]. It is proposed for its specific application to water supply systems.

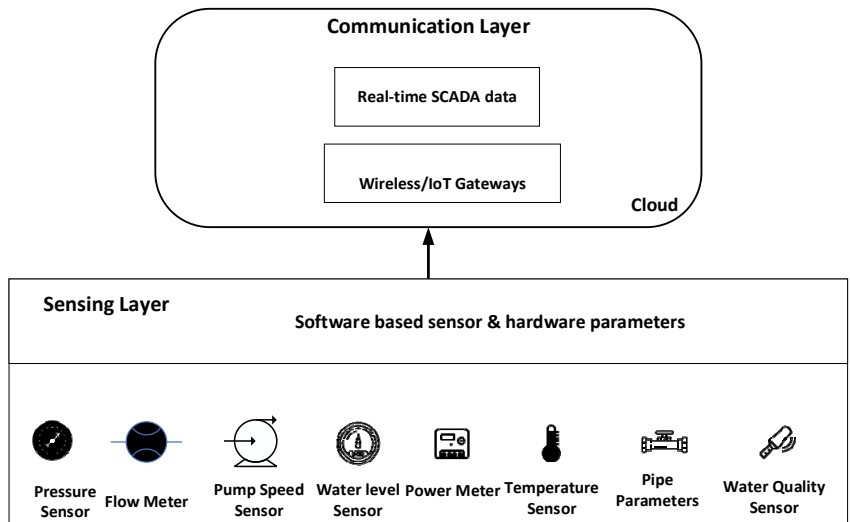

**Figure 2.** Communication and Sensing Layers.

Based on the literature review, a framework for the intelligent water network has been developed and depicted in Figure 3.

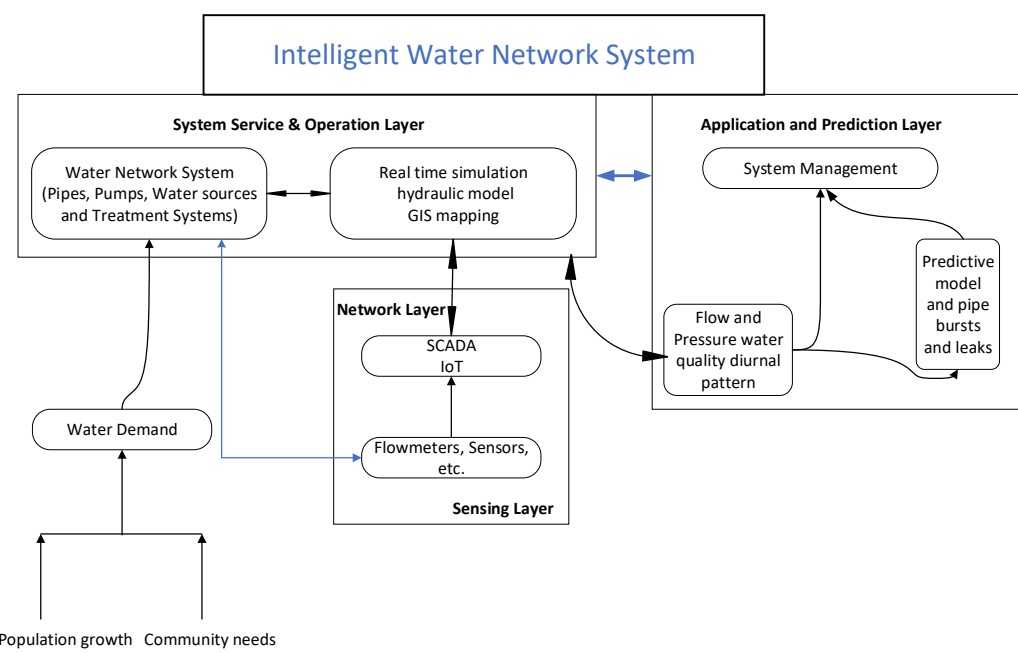

**Figure 3.** Framework for an Intelligent Water Network System.

An IoT framework can include several layers [124]; in Figure 3, four layers are proposed for developing the framework for an IWN, which are: (i) the sensing layer, (ii) communication layer, (iii) water system and operation layer, and (iv) application and prediction layer. Sensors and flow meters send information about flow, pressure, and water quality parameters to SCADA. Optimal allocation of pressure sensors and flow meters will be based on the local topography, size of the water supply system, and fluctuations in water quality over time due to ageing infrastructure, based on water quality historical data.

In a water system, the flow meters, pressure sensors and other monitoring devices are connected through the SCADA system to the data analysis centre. Flow and presure sensors' data are used in calibrating the hydraulic mode and comparison with real time simulation of the water network. GIS mapping can provide the information on the water

network. The prediction models provide warnings about the network conditions, which can be used for the overall asset planning, maintenance scheduling and overall operation.

The flow and pressure data analysis using AI models identifies leaks and bursts in the system.

## 6. SWOT Analysis of Water Systems

The SWOT analysis of an IWN has been shown in Figure 4. There are various opportunities in developing an intelligent water network, which are related to system efficiency, informed decision making with better data, and better prediction of leaks and busts resulting in minimisation of loss revenue including water conservation. However, the cost of developing IWNs and the technical manpower to manage these systems and assess any information due to faulty sensors in decision making can be possible threats to their implementation.

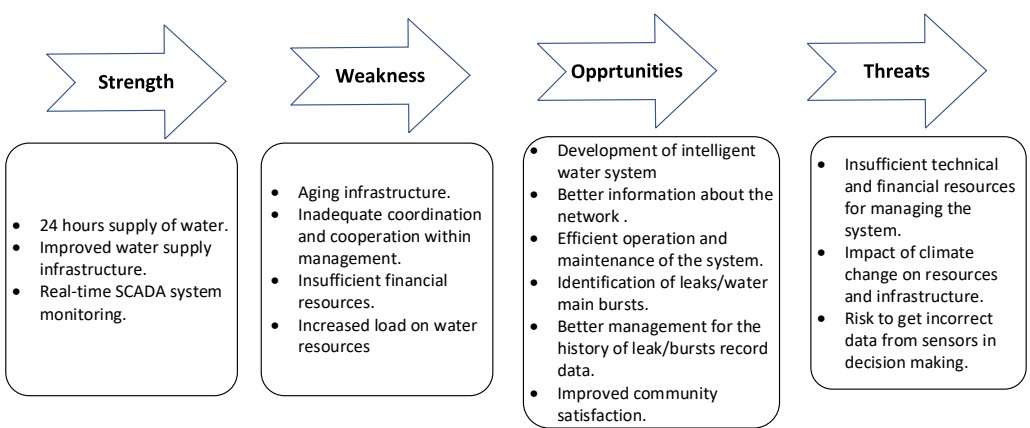

**Figure 4.** SWOT Analysis of Water System.

## 7. Conclusions

IWN systems are being implemented to address operational and management challenges faced by water utilities. These authorities incorporate smart technologies and analysis and modelling techniques in existing and new systems. These technologies play a significant role in enhancing system operational and management efficiency with live system data for informed and quick decision making; better and timely management of pipe leaks and bursts, allowing utilities to take a more proactive approach to ageing infrastructure; reliable water quality management; improved customer experience; optimal electricity consumption; reliable remote operation, including efficient manpower management. High-quality, efficient pressure sensors, water quality sensors, high-resolution flow meters, efficient data analysis and modelling tools, and predictive models are employed in intelligent water systems. AI and other analytical techniques are integral components of such networks employed for live data analysis, such as hydraulic simulation and predictive modelling for asset management. Water utilities can ensure enhanced customer satisfaction, and ensure sustainable and reliable water supply with such systems. Effective preventive maintenance of high-risk pipelines will even more reduce economic losses and reduce water quality issues with enhanced asset management.

A framework for the comprehensive development of an intelligent water network is presented, which includes input data on water demand, system characteristics for pipes, pumps, water sources and water treatment, sensors and flow meters, SCADA, system predictive models, tools for real-time simulation and hydraulic modelling, and output data for operators and managers for well-informed decision making. It is hoped that the proposed framework will help water professionals across globe in developing IWNs.

**Author Contributions:** K.J., A.K.S. and R.v.S. (authors) have worked on this manuscript together. All authors provided substantial contributions to the conception design of the work; the acquisition analysis, interpretation of data for the work; and drafting the work including revising it critically for important intellectual content. All authors have read and agreed to the published version of the manuscript.

**Funding:** This research is funded by Victoria University, Melbourne Australia and Greater Western Water, Melbourne, Australia.

**Informed Consent Statement:** Not applicable.

**Conflicts of Interest:** The authors declare no conflict of interest.

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
