# Peer review of "Development of an Intelligent Urban Water Network System"

_water, doi:10.3390/w14091320_

Round 1

Reviewer 1 Report

Development of an Intelligent Urban Water Network System

The manuscript is interesting although the authors should improve deeply the structure and writing.

Really, the English grammar is poor and it is further of a research paper.

Abstract. It should be rewritten and the author should establish a summarize of their review, not only sentences, which are not connected between them

The manuscript tried to develop an time evolution of the water management. It is interesting but the authors have to improve the writing and structure of the ideas. Maybe, the authors should do a key figure in which they can show this evolution. Besides, they should add a table where the different topics and challenges are shown

They speak about the interconnection of SCADA with smart cities, maybe, they have to present the current technologies. The authors can support in this research “ Smart water management towards future water sustainable networks. Water12(1), 58.”

They established the need to develop a smart water management but they did not connect this smart management to reach a best sustainable water systems. In my opinion, the authors should explore this issue and they have to join with the smart management. Some of the references, which can help authors are:

Objectives, Keys and Results in the Water Networks to Reach the Sustainable Development Goals. Water, 13(9), 1268.

Barriers to extending piped water distribution networks: the case of Ekiti State, Nigeria. Utilities Policy63, 100983.

Monitoring inequality in water access: Challenges for the 2030 Agenda for Sustainable Development. Science of the Total Environment727, 138746.

 Sustainable management of water resources: agricultural sector and environmental protection. Sustainable management of water resources: agricultural sector and environmental protection, 1-7.

Eco-efficiency analysis of integrated grey and black water management systems. Resources, Conservation and Recycling172, 105681.

Rethinking the economics of water: An assessment. Oxford Review of Economic Policy36(1), 1-23.

Some sub-sections (example 2.1.3 ) should be joined with others. The authors cannot do sub-section with a paragraph only.

Section 4,5,6 and 7 should be restructured. The authors write about topics but they did not connect between them. This is not a book and therefore, the authors should do a flowchart where they show the review line and the connection between different topics.

The authors should develop a swot matrix to show the new challenge in water sector

The authors should show the new futures lines in the conclusion

Author Response

Dear Editor

Authors would like to thank reviewer 1 for their valuable comment, which have helped in improving the structure and quality of the paper. The response to reviewers’ comments and the description of the changes made to incorporate comments are provided in the attached document.

Reviewer 2 Report

Dear Authors,

Following are my comments:  

1) Introduction - Needs to be restructured. It needs to have few paragraphs stating what is the issue, why is it an issue, current state of the issue and structure of the paper. Also section 1.1 is irrelevant for readers. 

2) There are many sentences which has similar meaning or have been repeated throughout the article. Please remove them. 

3) Section 2.1 Characteristics of IWN - I do not think that water quality and pressure sensors are the characteristics.

4)  Following articles are relevant to your research: a) An Internet of Things Approach for Water Efficiency: A Case Study of the Beverage Factory b) Improving Water Efficiency in the Beverage Industry with the Internet of Things

5) The authors have used INR as currency. Please convert it either to Euros or US dollars.

Author Response

Dear Editor

Authors would like to thank reviewer 2 for their valuable comment, which have helped in improving the structure and quality of the paper. The response to reviewers’ comments and the description of the changes made to incorporate comments are provided in the attached document.

Round 2

Reviewer 1 Report

The authors clarified the different suggestions

Reviewer 2 Report

The revised manuscript reads well and have addressed all my suggested comments. The only minor change is to the author contributions section which currently is not in line with mdpi format. Please refer to other mdpi papers.